

# Heme oxygenase-1: potential therapeutic targets for periodontitis

Weiwei Lv[1,2,*], Shichen Hu[1,2,*], Fei Yang[1,2], Dong Lin[1,2], Haodong Zou[1,2], Wanyan Zhang[1,2], Qin Yang[3], Lihua Li[1,2], Xiaowen Chen[4] and Yan Wu[1,2]

[1] Department of Stomatology, Affiliated Hospital of North Sichuan Medical College, Nanchong, Sichuan, China
[2] Department of Stomatology, North Sichuan Medical College, Nanchong, Sichuan, China
[3] School of Pharmacy, North Sichuan Medical College, Nanchong, Sichuan, China
[4] School of Medical Imaging, North Sichuan Medical College, Nanchong, Sichuan, China
* These authors contributed equally to this work.

## ABSTRACT

Periodontitis is one of the most prevalent inflammatory disease worldwide, which affects 11% of the global population and is a major cause of tooth loss. Recently, oxidative stress (OS) has been found to be the pivital pathophysiological mechanism of periodontitis, and overactivated OS will lead to inflammation, apoptosis, pyroptosis and alveolar bone resorption. Interestingly, heme oxygenase-1 (HO-1), a rate-limiting enzyme in heme degradation, can exert antioxidant activites through its products—carbon monoxide (CO), $Fe^{2+}$, biliverdin and bilirubin in the inflammatory microenvironment, thus exhibiting anti-inflammatory, anti-apoptotic, anti-pyroptosis and bone homeostasis-regulating properties. In this review, particular focus is given to the role of HO-1 in periodontitis, including the spatial-temporal expression in periodental tissues and pathophysiological mechanisms of HO-1 in periodontitis, as well as the current therapeutic applications of HO-1 targeted drugs for periodontitis. This review aims to elucidate the potential applications of various HO-1 targeted drug therapy in the management of periodontitis, investigate the influence of diverse functional groups on HO-1 and periodontitis, and pave the way for the development of a new generation of therapeutics that will benefit patients suffering from periodontitis.

## INTRODUCTION

Periodontitis is recognized as one of the most prevalent chronic inflammatory diseases affecting the oral cavity in humans and is the primary cause of tooth loss among adults. Severe periodontitis exerts a significant socio-economic impact on a global scale. Recent estimates suggest that the economic burden of severe periodontitis amounts to approximately $54 billion annually. Moreover, as the population ages, both the incidence of periodontitis and its associated burden are projected to increase each year (*Tonetti et al., 2017*).

Corresponding author
Yan Wu, yanwu@nsmc.edu.cn

Nevertheless, owing to the intricate nature of the pathogenesis of periodontitis, current therapeutic interventions remain insufficient to achieve complete resolution in the majority of cases. When exploring the various factors involved in the pathogenesis of periodontitis, it was found that although plaque biofilm is the initiator of periodontitis, other factors (*e.g.*, genetics, obesity, systemic systemic diseases, smoking, *etc.*) also have an impact on the inflammatory response of periodontal tissues (*Genco & Borgnakke, 2013*). Epigenetics can regulate gene expression without altering the DNA sequence, thereby affecting the inflammatory process by suppressing or overstimulating genes. For example, LncRNAZFY-AS1 can inhibit periodontitis and reduce oxidative stress by regulating microRNA-129-5p to promote DEAD-Box helicase 3 X-Link (*Cheng et al., 2022*). Interestingly, the Third National Health and Nutrition Examination Survey in the United States reported a significant association between obesity and the incidence of periodontitis in young people (*Al-Zahrani, Bissada & Borawskit, 2003*), and previous studies found that exercises could reduce systemic reactive oxygen species (ROS) and effectively inhibit gingival oxidative stress caused by obesity (*Azuma et al., 2011*). On the other hand, chronic periodontitis is the sixth complication of diabetes mellitus, and local and systemic oxidative damage exacerbated the severity of periodontitis in diabetes in a rat mode (*Li et al., 2018*). A comprehensive analysis of data from the 2016–2018 National Health and Nutrition Examination Survey in South Korea revealed that smokers exhibited a significantly higher risk of periodontitis compared to nonsmokers (*Sim et al., 2023*). Research indicated that smoking led to heightened total oxidative stress and reduced antioxidant capacity within periodontal tissue (*Lütfioğlu et al., 2021*).

Consequently, numerous factors indicate that oxidative stress, particularly ROS, plays a pivotal role in the pathogenesis of periodontitis (*Nazir, 2017*; *Xie et al., 2023*). Upon stimulation by bacterial lipopolysaccharide (LPS), various inflammatory cells aggregate into the periodontal tissues, of which neutrophils produce ROS to eliminate pathogens. However, prolonged stimulation by dental plaque biofilm leads to excessive ROS production, causing toxic effects on cellular macromolecules and mitochondrial damage. Then, mitochondrial damage further disrupts cellular energy metabolism, ultimately leading to cell death. On the other hand, excess ROS as well as insufficient antioxidant capacity of the body contribute to the disruption of the balance between oxidation and antioxidation in periodontal tissues. Cell membranes contain large amounts of polyunsaturated fatty acids esterified on phospholipids and free cholesterol, and these lipids are the main targets of ROS attack (*Kumar et al., 2017*). Additionally, it has been found that oxidative stress may indirectly affect oral bone tissue by generating oxidized fatty acids, which activate adipogenesis and inhibit osteoblastogenesis while directly impact osteoclasts. Detaily, ROS upregulates the expression of receptor activator of nuclear factor κB ligand (RANKL) and tumor necrosis factor-α (TNF-α), which are considered as important driving factors for osteoclast formation and bone resorption activity (*Żukowski, Maciejczyk & Waszkiel, 2018*).

Oxidative stress is characterized by an imbalance between oxidative and antioxidative processes within the body. The assessment indices encompass superoxide dismutase (SOD), catalase (CAT), malondialdehyde (MDA), glutathione (GSH), 8-hydroxylated

deoxyguanosine (8-OHdG) and ROS, among others, which collectively reflect the extent of oxidative stress and associated bodily damage. Randomized controlled trials investigating antioxidant capacity have revealed elevated levels of oxidative stress markers and diminished antioxidant status in the saliva or blood of periodontitis patients, in comparison to a control group (*Baltacıoğlu et al., 2014*). Data from the Third National Health and Nutrition Examination Survey in the United States demonstrated a negative correlation between indicators of serum antioxidant status—including serum vitamin C, bilirubin and total antioxidant capacity (TAOC)—and the prevalence of periodontitis (*Chapple, Milward & Dietrich, 2007*). Antioxidant enzymes, such as SOD, CAT and glutathione peroxidase (GPx), exhibited diminished activity levels in individuals diagnosed with periodontitis (*Huang et al., 2014*). MDA, a byproduct of membrane lipid peroxidation, was found to be significantly elevated in the saliva of patients suffering from periodontitis (*Cherian et al., 2019*). GSH, a critical antioxidant engaged in scavenging free radicals and participating in various antioxidative reactions, was observed to be significantly reduced in the serum levels of patients diagnosed with periodontitis (*Biju et al., 2014*). 8-OHdG, a significant indicator of oxidative DNA damage, demonstrated a substantial positive correlation with the severity of periodontitis (*Varghese et al., 2020*). Furthermore, Gustafsson reported that peripheral blood neutrophils exhibited elevated production of ROS in patients with chronic or aggressive periodontitis, indicating heightened neutrophil reactivity among these patients (*Gustafsson & Asman, 1996*). Collectively, these findings suggest that the inhibition of ROS levels may exert a beneficial effect on mitigating periodontal tissue destruction and suppressing alveolar bone resorption.

Current studies have revealed that over-activated oxidative stress would induce several cytokines and signal pathways to protect the body against ROS damage, including transcription factors (TFs), activator protein-1 (AP-1), NF-κB, mitogen-activated protein kinase (MAPK) and nuclear factor erythrocyte 2-related factor 2 (Nrf2). Interestingly, most of these cytokines and signal pathways could activate the heme oxygenase (HO) system to play an important antioxidant role to protect periodontal cells (*Huang et al., 2021a*; *Zhou et al., 2018*). The HO system was first identified in 1968 and recognized as a coupled NADPH-dependent microsomal oxygenase system (*Tenhunen, Marver & Schmid, 1968*). The HO system consist of three isozymes, including HO-1, HO-2 and HO-3 (*McCoubrey, Huang & Maines, 1997*). Of these, HO-2 is constitutively expressed at high levels in brain, testis, or endothelial cells, whereas HO-3 was only rarely observed in rat astrocytes (*Fernández-Fierro et al., 2020*). However, both HO-2 and HO-3 are constitutively expressed, undergo intense expression independent of external stimuli, and function primarily in normal heme capture and metabolism (*Naito et al., 2011*; *Donnelly & Barnes, 2001*). In addition, HO-1, a heat shock protein 32 (HSP32), is a stress-inducible isoform encoded by the human HMOX1 gene, and widely expressed in various tissues (*Leal & Carvalho, 2022*). Genetic defects in HO-1 can cause endothelial cell damage, anemia, and aberrant tissue iron accumulation stimulated by oxidative stress (*Yachie et al., 1999*). Moreover, HO-1 was found to have antioxidant effects through increasing the activities of antioxidant enzymes such as GPx, SOD and CAT, while decreasing the

expression of MDA, a marker of oxidative stress (*Rizzardini et al., 1994*; *Rawlinson et al., 1998*; *Ding et al., 2016*; *Wei et al., 2015*). Consequently, HO-1 has been recognized as an anti-inflammatory mediator involved in several inflammatory diseases such as hepatitis, neuritis and nephritis (*Ma et al., 2024a*; *Tao et al., 2024*; *Berköz, Yiğit & Krośniak, 2023*). We hypothesized that regulating the expression and activity of HO-1 may provide new ideas for the treatment of periodontitis. This review will discuss the current research progress on HO-1 and focus on its mechanism of action in different periodontitis phenotypes, as well as its upstream and downstream related targets and potential targeted therapeutic agents. Ultimately, we will use these interventions to target HO-1 in the treatment of periodontitis.

## SURVEY METHODOLOGY

We conducted a comprehensive literature review on HO-1 in the context of periodontitis through searches on various platforms, including the Web, Google Scholar, and PubMed. The search terms employed included "HO-1", "oxidative stress", "periodontitis" and "ROS". This was achieved by intersecting these descriptors utilizing Boolean operators, specifically "OR" and "AND". Primarily, we included pertinent articles published from 2010 until July 2024 or earlier. Articles pertaining to cancer, non-oxidative stress-related diseases, viral infections, and those unrelated to periodontitis and HO-1 were systematically excluded. In summarizing the degradation products of HO-1, we incorporated key search terms such as "heme", "carbon monoxide", "$Fe^{2+}$", "biliverdin" and "bilirubin". To elucidate the molecular mechanisms of periodontitis associated with HO-1, we also integrated terms such as "PI3K/Akt", "Nrf2", "HMGB1", "MAPK" and "NF-κB". In organizing the literature concerning the biological behaviors associated with periodontitis, we further included keywords such as "inflammation", "bone resorption", "apoptosis" and "pyroptosis". Additionally, we incorporated terms such as "polyphenols", "terpenoids", "isothiocyanates", "saponins" and "alkaloids" as significant search keywords for HO-1-targeted therapies related to the treatment of periodontitis. In summary, this review identified a total of 215 pertinent research articles that examined the influence of HO-1 on the biological behaviors associated with periodontitis, as well as the roles of various functional groups in this context.

### HO-1 metabolite

Basically, HO-1 exerts its effects through the degradation of heme. Heme (iron protoporphyrin IX), an iron porphyrin compound, is a cofactor of hemoglobin, myoglobin, cytochromes, peroxidases and catalase (*Wagener et al., 2001*). Heme is involved in a variety of biological processes, such as oxygen transport, signal transduction, peroxide metabolism and energy production (*Chiabrando et al., 2014*). The metabolism of periodontal cells cannot be achieved without the help of heme, but excessive free heme may activate Toll-like receptor 4 (TLR4) to produce deleterious effects by catalyzing the production of ROS, leading to oxidative stress and inducing cellular damage (*Ryter & Tyrrell, 2000*; *Janciauskiene, Vijayan & Immenschuh, 2020*). Furthermore, *Porphyromonas gingivalis*, the primary causative agent of periodontitis, is unable to proliferate without the

presence of heme (*Gao et al., 2018*). Consequently, high expression of HO-1 under the sustained effect of periodontal inflammation can degrade free heme to avoid cellular damage and act as an antimicrobial agent to a certain extent. Simultaneously, in response to periodontal inflammation, HO-1 forms a complex with heme and NADPH-cytochrome P450 reductase (*Yoshida & Kikuchi, 1978*). NADPH serves as an electron donor, while molecular oxygen binds to the complex, resulting in the production of CO, $Fe^{2+}$ and biliverdin (*Yoshida, Noguchi & Kikuchi, 1980*). Subsequently, biliverdin is enzymatically converted to bilirubin by biliverdin reductase (*Wang & de Montellano, 2003*). HO-1 exerts its antioxidant effects *via* the degradation of these byproducts, thereby providing protective benefits to periodontal cells.

### Carbon monoxide

Carbon monoxide (CO), a small molecule gas, contributes at least 86% of the endogenous carbon monoxide (*Ryter, Alam & Choi, 2006*). Low concentrations of CO have protective, anti-inflammatory, antioxidant and antibacterial properties (*Di Pietro et al., 2020*). MAPK are important targets of CO, regulating various important physiological and pathological processes such as cell growth, differentiation, environmental adaptation and inflammatory responses. In rheumatoid arthritis studies, CO was found to exert antioxidant properties by reducing ROS activity through inhibition of NF-κB expression rather than the MAPK pathway (*Ryter, Ma & Choi, 2018*). However, in the study on gingival fibroblasts, it was found that CO only activated P38 MAPK and had no effect on JNK and ERK expression (*Song et al., 2011*). Further research is needed to confirm whether CO affects the expression of MAPKs in other periodontal-related cells. Additionally, the anti-inflammatory effects of CO are related to the regulation of TLR4. CO can inhibit the expression of inflammatory factors such as IL-1β, $PGE_2$ and inducible nitric oxide synthase(iNOS) in periodontitis (*Song et al., 2017*; *Choi et al., 2021*). It was found that in diabetic periodontitis, CO could mediate the RAGE/NF-κB pathway to suppress periodontal tissue inflammation (*Tian et al., 2024*). Reports indicate that CO inhibits osteoclast formation while simultaneously promoting osteogenic differentiation in the context of periodontitis (*Song et al., 2017*). The Nrf2/HO-1 pathway is another target of CO, which exerts antioxidant effects by activating antioxidant enzymes such as SOD, CAT and GPx (*Di Pietro et al., 2020*). Moreover, transcription activation factor-3 (STAT-3), phosphatidylinositol 3-kinase/Akt (PI3K/Akt) and HIF-1, are also the targets of CO, which are involved in the regulation of cell protection (*Ryter, Ma & Choi, 2018*).

### $Fe^{2+}$

Free ferrous iron exhibits elevated expression levels during the pathological progression of periodontitis and plays a crucial role in essential physiological activities, including DNA synthesis, ATP synthesis, and oxygen transport (*Chen et al., 2022a*; *Frey & Reed, 2012*). However, when excessive storage of ferrous ions occurs, resulting in iron overload, it may initiate the Fenton reaction, catalyzing the generation of ROS and leading to lipid peroxidation and subsequent oxidative damage to tissues (*Rehncrona et al., 1982*).

HO-1 plays an important role in cytoprotection, effectively helping cells to resist damage. However, concurrently, $Fe^{2+}$, another metabolite of HO-1, has the potential to contribute to the generation of free radicals. This apparent contradiction has long perplexed researchers and has stimulated related studies to explore this phenomenon more deeply. It has been proposed that this may be closely linked to the observation that $Fe^{2+}$ produced by HO-1 activity is associated with elevated levels of ferritin and transferrin (*Lanceta et al., 2013*; *Paiva et al., 2012*). These proteins are integral to the pathology of periodontitis, particularly in the regulation of iron metabolism and the mitigation of oxidative stress. The ferroxidase center located within the ferritin heavy chain may further facilitate the oxidation of $Fe^{2+}$ to the relatively less reactive ferric iron ($Fe^{3+}$) (*Bou-Abdallah, 2010*; *Lawson et al., 1989*). It has been proposed that HO-1 deficiency may result in disruptions in iron metabolism, consequently leading to reduced transferrin levels (*Kartikasari et al., 2009*). However, following 3 months of non-periodontal surgical treatment, serum transferrin levels increased significantly, indicating that these proteins are crucial in the management of periodontitis (*Shirmohamadi et al., 2016*). Furthermore, transferrin receptor 2 has been reported to effectively modulate the inflammatory response in periodontitis, thereby attenuating alveolar bone loss (*Lösser et al., 2024*). Thus, HO-1 not only safeguards cells from damage but may also mitigate alveolar bone loss induced by periodontitis by regulating the expression of transferrin and the ferritin heavy chain. Therefore, the regulatory relationship between $Fe^{2+}$ produced by HO-1 activity and transferrin as well as the ferritin heavy chain may be of significant importance in the treatment and prevention of periodontitis, warranting careful consideration.

### Biliverdin and bilirubin

Biliverdin is produced by HO-1-catalyzed degradation of heme. It can be converted to bilirubin by biliverdin reductase and nicotinamide adenine dinucleotide phosphate (NADPH) (*Zhu et al., 2011*). Previously, bilirubin was thought to be a hazardous waste product that could lead to liver disease or neonatal jaundice. However, a research in 1954 showed that bilirubin exhibited antioxidant properties that protected vitamin A and unsaturated fatty acids from oxidation (*Osiak et al., 2020*). Further studies have found that the antioxidant activity of bilirubin increased in the conditions of change from atmospheric oxygen concentration (20%) to tissue oxygen concentration (2%). Moreover, bilirubin could effectively scavenge single-linear oxygen molecules, disrupt free radical chain reactions, and act as a potent antioxidant, surpassing even the antioxidant alpha-tocopherol (*Stocker et al., 1987*). Low levels of hyperbilirubinemia may reduce the incidence of oxidative stress-related diseases such as cardiovascular disease, diabetes, obesity and metabolic syndrome. For example, it has been shown that topical application of bilirubin promoted healing of diabetic skin wounds by upregulating antioxidant status, promoting angiogenesis and collagen deposition (*Zhao et al., 2024*). In addition, bilirubin has shown some anti-inflammatory effects, and it was found that bilirubin treatment significantly reduced the expression levels of iNOS, cyclooxygenase-2 (COX-2) and IL-6, which in turn alleviated gastrointestinal inflammation (*Nakao et al., 2004*). Also, bilirubin treatment could reduce the levels of pro-apoptotic genes and improve the function and

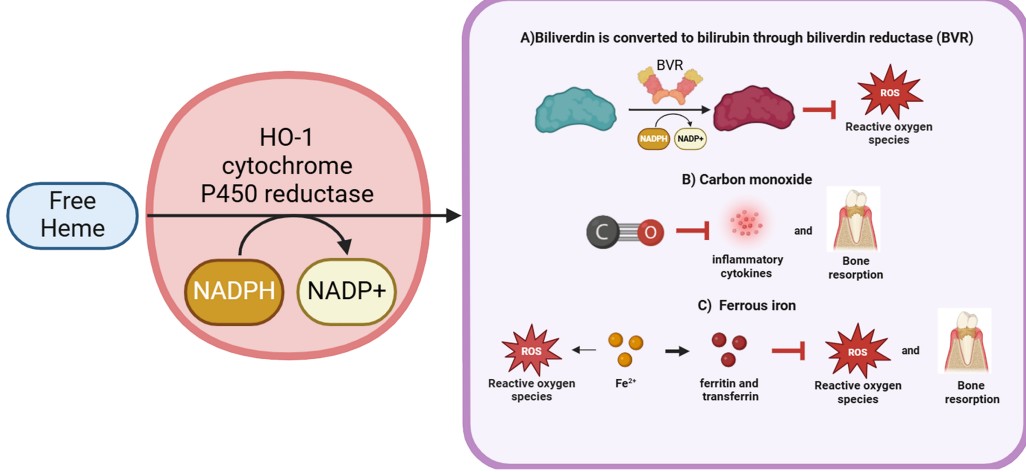

**Figure 1 The activity and final product generation of HO-1.** HO-1 is an enzyme that effectively degrades free heme, resulting in the production of biliverdin along with the release of carbon monoxide and $Fe^{2+}$. During the metabolism of biliverdin, it can be enzymatically converted to bilirubin by the action of biliverdin reductase. Notably, both biliverdin and bilirubin possess significant antioxidant properties, demonstrating the ability to effectively scavenge and neutralize ROS. Furthermore, carbon monoxide, a gaseous byproduct, inhibits the synthesis of inflammatory cytokines and partially attenuates alveolar bone resorption. Although $Fe^{2+}$ acts as a pro-oxidant, contributing to elevated ROS levels, HO-1, recognized as a potent antioxidant, facilitates the release of $Fe^{2+}$, which enhances the expression of ferritin and transferrin, thereby exerting antioxidant effects and regulating bone homeostasis. Thus, HO-1 plays a pivotal role in cytoprotection, with its mechanism of action primarily manifested through the regulation of inflammatory responses, enhancement of antioxidant capacity, and maintenance of bone homeostasis. HMGB1 is a potential downstream target of HO-1, which regulates periodontal inflammation by inhibiting HMGB1. Figure drawing was supported by Biorender (https://app.biorender.com/).

survival of allografts. Importantly, in treatments targeting periodontitis, researchers have found a negative correlation between bilirubin serum concentrations and the severity of periodontitis (*Chapple, Milward & Dietrich, 2007*). Bilirubin, derived from the reduction of biliverdin, is a potent antioxidant that effectively scavenges ROS, thereby preventing both protein and lipid peroxidation (*Zhou et al., 2021*) (Fig. 1).

## Expression levels of HO-1 in periodontal tissues

Many studies have shown that HO-1 was expressed at low levels in normal periodontal tissues, and only small amounts were secreted to maintain the balance between oxidative and antioxidant status within the cell (*Kim et al., 2009*; *Pi et al., 2010*; *Cho & Kim, 2013*). *Chen et al. (2021a)*, detected the expression of HO-1 in normal periodontal tissues of young and aged mice, and found that the expression of HO-1 was significantly higher in aged mice compared to young mice, which suggested that the expression of HO-1 may be related to the aging. Additionally, HO-1 can also be induced in the periodontal tissues by various stimuli, including oxidative stress, inflammation, high oxygen, hypoxia and tissue injury (*Hanselmann, Mauch & Werner, 2001*). Infected periodontal tissues may secrete HO-1, which participates in the protection of periodontal cells due to its anti-inflammatory, antioxidative and anti-apoptotic effects (*Park et al., 2011b*; *Liu et al., 2017*; *Zhao et al., 2021*). It has been reported that nicotine/LPS stimulation induced HO-1

expression in human gingival fibroblasts (*Chang et al., 2005*; *Qi et al., 2018*). Nicotine, LPS, IL-1β and TNF-α can also promote the secretion of HO-1 in periodontal ligament cells (*Kim et al., 2009*; *Pi et al., 2010*). Macrophages, derived from monocytes migrating to tissues and differentiating into mature cells, play important roles as immune accessory cells in both normal physiological processes and pathological processes. In inflamed periodontal sites, macrophages account for approximately 5–30% of infiltrating cells. LPS from *Porphyromonas gingivalis* or *Prevotella intermedia* can induce the secretion of HO-1 in RAW264. 7 macrophages (*Cho & Kim, 2013*; *Park et al., 2011b*). Furthermore, human periodontal ligament stem cells (HPDLSCs) are another important cellular component in periodontal ligaments. They possess self-renewal and multidirectional differentiation potential, maintain periodontal homeostasis and participate in periodontal regeneration. $H_2O_2$ stimulated HO-1 expression in HPDLSCs (*Liu et al., 2017*). LPS promoted HO-1 expression in human oral gingival epithelial keratinocytes (*Hagiu et al., 2020*). Substance P, as a pro-inflammatory peptide involved in inflammation and immune response, increased HO-1 expression in periodontal ligament cells (*Lee et al., 2007*). Milk can promote the expression of inflammatory factors and matrix metalloproteinases (MMPs), while also inducing HO-1 secretion in human periodontal ligament cells (*Choi et al., 2015c*). In an *in vivo* experiment using a mouse periodontitis model, HO-1 expression was upregulated (*Kataoka et al., 2016*). However, it is still unknown which tissue is the main source of HO-1 production in periodontal tissues. Interestingly, not all studies have demonstrated high expression of HO-1 in inflamed periodontal tissues. In human gingival fibroblasts, one report indicated that LPS reduces HO-1 expression (*Huang et al., 2022*). In a simulated diabetic periodontal cell model, the expression of HO-1 in HPDLSCs was decreased (*Mohamed Abdelgawad et al., 2021*). The differential expression of HO-1 in various cell types and species may be attributed to polymorphic segments within the HO-1 promoter gene (*Taha et al., 2010*). It has been shown that longer polymorphic fragment sequences are associated with reduced HO-1 expression and reduced resistance to oxidative stress (*Loboda et al., 2008*). Furthermore, no researchers have yet examined the changes in HO-1 expression between normal and inflamed periodontal tissues in humans, which needs to be further investigated.

## HO-1 and periodontitis

### Pathophysiological mechanism

The development of periodontitis involves several factors, including inflammation, oxidative stress, apoptosis, pyroptosis and alveolar bone resorption.

### Inflammation

When the host is infected with periodontal pathogens, the body activates the immune system to generate a defense response. Periodontal tissues recruit large numbers of neutrophilic polymorphonuclear leukocytes that produce the release of proinflammatory factors such as iNOS, IL-1β, IL-6 and IL-18 to destroy periodontal pathogens (*Sculley & Langley-Evans, 2002*; *Zhang et al., 2021*; *Huang et al., 2020*). On the one hand, NO is usually produced through the metabolism of the L-arginine catalyzed by nitric oxide

synthase (NOS) enzymes. To date, two isoforms of constitutive expression (nNOS and eNOS) have been identified, whereas the iNOS isoform is induced only in response to infection, inflammation, or trauma (*Nathan & Xie, 1994*). The HO-1 inhibitor tin protoporphyrin IX (SnPP) reduces the expression of ROS, which in turn inhibits the expression of iNOS in periodontitis (*Choi et al., 2013*). Studies have reported a shift in iNOS from NO production to ROS production, and iNOS is regulated by ROS (*Sun, Druhan & Zweier, 2010*).

*Oxidative stress and apoptosis*
ROS is a general term for oxygen radicals and peroxides, which have important roles in intracellular signaling and antimicrobial activity. Excess ROS can exert cytotoxicity and induce lipid peroxidation. MDA is used as a reliable indicator of oxidative damage in periodontitis. The results of the study showed that lipid peroxidation levels were higher in periodontitis patients than in periodontally healthy individuals, and MDA levels were closely associated with periodontal tissue inflammation and supporting tissue destruction (*Mohideen et al., 2023*). The construction of a mouse model of periodontitis found that high expression of HO-1 reduced the expression of ROS and oxidative stress damage marker MDA (*Zhao et al., 2021*). In addition, excessive ROS can induce oxidative DNA damage. For example, patients with chronic periodontitis have significantly elevated levels of 8-OHdG, a marker of oxidative DNA damage (*Chen et al., 2019a*). This oxidative DNA damage can interfere with cell cycle progression, ultimately leading to apoptosis of periodontal cells (*Chang et al., 2013*; *Yu et al., 2012*). *Jiang et al. (2022)* found that activation of the Nrf2/HO-1 pathway inhibited the expression of apoptosis markers Caspase3/9, which, in turn, produced a protective effect on periodontal cells.

*Pyroptosis*
On the other hand, NLRP3 seems to be associated with the production and maturation of pro-inflammatory cytokines (*e.g.*, IL-1β and IL-18) as well as the onset of pyroptosis, leading to chronic inflammation. However, the conversion of IL-1β and IL-18 to an active state as well as the promotion of highly inflammatory forms of programmed cell death (i.e., pyroptosis) are not possible without the help of caspase-1 (*Huang et al., 2020*). Thus, there is evidence that ROS may promote periodontal tissue inflammation by inducing NLRP3-caspase-1-IL-1β (*Yoon et al., 2018*). Recent studies have found that HO-1 has a regulatory effect on cellular pyroptosis. Unlike apoptosis, pyroptosis is a mechanism of programmed cell death of inflammatory cells mediated by caspase-1/-4/-5/-11, which is characterized by cell lysis and release of proinflammatory factors. HO-1 inhibits NLRP3 inflammasome and subsequently suppresses Caspase-1 activity, thereby preventing pyroptosis in gingival fibroblasts (*Huang et al., 2020*).

*Alveolar bone resorption*
*Kim et al. (2005)* demonstrated that osteoclast differentiation could not be achieved without the help of nuclear factor of activated T-cell c1 (NFATc1). RANKL-mediated ROS activation of NFATc1 promotes osteoclast differentiation (*AlQranei et al., 2020*). *Liu et al. (2019a)* further confirmed that activation of HO-1 reduces ROS to inhibit NFATc1, which

in turn inhibited bone destruction in a variety of diseases including periodontitis. Other studies have confirmed that both diabetes and periodontitis can undergo varying degrees of oxidative stress damage, and that promoting HO-1 expression results in decreased ROS levels, reduced bone resorption, wound healing, and amelioration of diabetic damage to periodontal tissues (*Liu et al., 2021*).

### Molecular mechanism

Studies have shown that pathological processes such as inflammation, oxidative stress, apoptosis, pyroptosis and alveolar bone resorption in the treatment of periodontitis are closely related to the regulation of HO-1. Therefore, HO-1 has gradually become an important hub in the treatment of periodontitis, attracting great attention from scholars. It has been shown that inducers can regulate HO-1 expression through multiple signaling pathways, which include NF-κB, PI3K-Akt, p38 MAPK and Nrf2, *etc.*, (*Park et al., 2011a*; *Jin et al., 2012*; *Du et al., 2022*).

The NF-κB signaling pathway is activated by ROS, culminating in the upregulation of pro-inflammatory cytokines and chemokines, which ultimately leads to the destruction of periodontal tissues (*Özcan et al., 2017*). Moreover, numerous studies have demonstrated that CO, a catabolic byproduct of HO-1, can alleviate periodontal inflammation by inhibiting the NF-κB signaling pathway (*Song et al., 2017*; *Choi et al., 2021*, *2022*). Additionally, the PI3K/Akt pathway is pivotal in the progression of periodontitis, influencing cell proliferation, apoptosis, cytokine secretion and osteoblast differentiation (*Liu et al., 2019b*). Notably, researchers have revealed that the activation of the PI3K/AKT/HO-1 signaling cascade inhibits periodontal inflammation and imparts significant anti-oxidative stress properties (*Park et al., 2011a*). On the other hand, the family of MAPKs (including ERK1/2, JNK, and p38) mediates fundamental biological processes and cellular responses in response to hormones, growth factors, cytokines, bacterial antigens and environmental stresses (*Souza et al., 2012*). JNK has been shown to exert anti-apoptotic effects in response to bacterial invasion, with its activation stimulating the expression of genes associated with resistance to oxidative stress and apoptosis (*Wang et al., 2015*). Furthermore, several studies have indicated that the activation of specific signaling cascades, such as ERK/HO-1 and JNK/HO-1, can induce an anti-oxidative stress response, thereby effectively suppressing periodontal inflammation (*Park et al., 2011a*; *Jeong et al., 2010*).

Nrf2 serves as the principal activator of HO-1. In its resting state, Nrf2 associates with kelch-like ECH-associated protein 1 (Keap1), which is sequestered in the cytoplasm and subjected to degradation *via* ubiquitination. Under oxidative stress conditions, Keap1 dissociates from Nrf2, resulting in the translocation of Nrf2 to the nucleus, the dissociation of BTB and CNC homology 1 (Bach1) from the HO-1 promoter, and subsequently, the activation of HMOX1 gene expression (which encodes HO-1) (*Zhou et al., 2021*). For instance, metformin induces the dissociation of Keap1 from Nrf2, thereby enhancing HO-1 expression and exerting a therapeutic effect on diabetic periodontitis (*Mohamed Abdelgawad et al., 2021*). Furthermore, research has demonstrated that inhibition of the DNA-binding activity of Bach1, coupled with the promotion of its degradation, can lead to

the upregulation of HO-1 expression, thereby facilitating periodontal tissue regeneration (*Yuan et al., 2024*). Beyond the Nrf2/HO-1 signaling mechanism, various other factors influencing HO-1 expression have been extensively investigated. For example, ginsenosides promote HO-1 expression *via* epidermal growth factor receptor (EGFR), and knockdown of EGFR leads to decreased HO-1 expression, which in turn reverses the anti-inflammatory and osteogenic effects of periodontal tissues (*Kim et al., 2021*). Additionally, microRNAs are implicated in the regulation of HMOX1 and its upstream genes (*e.g.*, Nrf2, Keap1, Bach1, *etc.*), offering new avenues for research into the regulation of HO-1.

HO-1 may also play a role in regulating the expression of high mobility group box 1 (HMGB1), a highly conserved nuclear protein implicated in the development of various inflammatory diseases, including periodontitis. The release of HMGB1 may be induced under conditions of oxidative stress. However, the activation of HO-1 contributes to the suppression of HMGB1 expression (*Yao et al., 2022*). In a murine model of neuropathic pain, activation of HO-1 was found to inhibit HMGB1 expression, consequently reducing the levels of pro-inflammatory factors (*Chen et al., 2019b*). *Ha et al. (2012)* demonstrated that the HO-1 inhibitor zinc protoporphyrin (ZnPP) exacerbates brain injury by inducing the expression of HMGB1. Additionally, elevated levels of HMGB1 were detected in the gingival sulcus fluid of patients with periodontitis, where HMGB1 induced the expression of pro-inflammatory factors in periodontal cells *via* activation of the NF-κB pathway (*Kim et al., 2010*; *Luo et al., 2011*). Consequently, HMGB1 may serve as a promising downstream target of HO-1 in the context of periodontitis.

In summary, the potential mechanisms underlying the role of HO-1 in periodontitis necessitate further in-depth investigation. By modulating HO-1 and its associated signaling pathways, novel strategies may emerge for the effective treatment of periodontitis. Currently, while gingival sulcus fluid and saliva are readily accessible in clinical settings, there are limited relevant clinical markers available for detecting the onset and progression of periodontitis. HMGB1, as a potential downstream target of HO-1, is anticipated to serve as a significant biomarker for the diagnosis and treatment of periodontitis. However, the majority of studies on periodontitis and biomarkers have been limited to cross-sectional investigations, primarily reporting correlations between biomarkers and periodontitis, while longitudinal studies examining these biomarkers throughout the progression of periodontitis are notably lacking. This limitation represents a significant challenge currently confronting periodontists. For example, it has been demonstrated that HMGB1 is highly expressed in the gingival sulcus fluid of individuals with periodontitis; however, the dynamics of HMGB1 during the progression of periodontitis and in response to HO-1-targeted therapy remain to be fully explored, which will be the focus of our future research. Further investigation into the potential mechanisms of action of HO-1 in periodontitis will pave the way for future personalized treatment strategies utilizing saliva and gingival sulcus fluid. Future studies should concentrate on effectively activating HO-1, mitigating its double-edged sword effect, and elucidating the specific roles of HO-1-related factors, such as HMGB1, in periodontitis. Furthermore, investigating the relationships between additional biomarkers and the

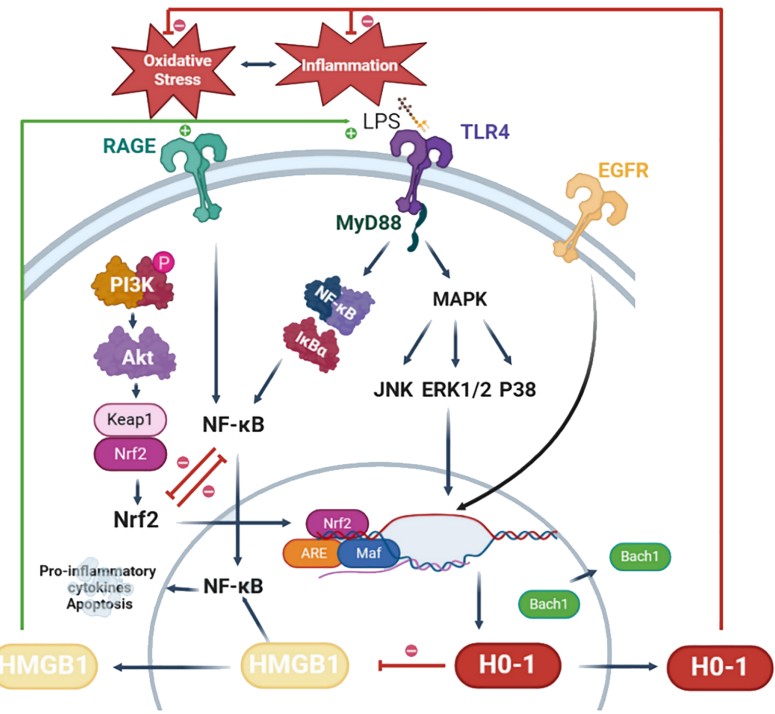

**Figure 2 Potential mechanisms of HO-1 in periodontitis.** LPS activates TLR4 signaling and initiates myeloid differentiation primary response protein 88-dependent (MyD88-dependent) pathway. MyD88 triggers the expression of NF-κB and MAPK leading to cell apoptosis and the production of pro-inflammatory cytokines. In particular, the MAPK pathway is involved in the regulation of HO-1. Phosphorylated expression of PI3K is increased in response to stimulation by anti-inflammatory drugs. Akt serves as a downstream target of PI3K. Akt stimulates the dissociation of Nrf2 and Keap1, translocating them into the nucleus. Bach1 bind to antioxidant response element (ARE) in the promoter region of the HO-1 gene, suppressing HO-1 expression. Nrf2 translocates into the nucleus, reducing Bach1's DNA-binding activity, inducing Bach1 nuclear export, and proteasome-dependent degradation. Nrf2 forms a complex with small Maf proteins (sMaf) and ARE. This complex facilitates the transcription of HO-1 and inhibits oxidative stress feedback. HMGB1 translocates from the nucleus to the cytoplasm, often undergoing post-translational modifications such as acetylation or phosphorylation. Through exocytosis, cells secrete HMGB1. Extracellularly, HMGB1 can activate and bind to its receptors, receptor for advanced glycation end products (RAGE), and TLR4, leading to NF-κB activation. This leads to the production and release of inflammatory cytokines and chemokines, triggering periodontal inflammation. Figure drawing was supported by Biorender (https://app.biorender.com/).

development of periodontitis aims to facilitate the development of more effective diagnostic methods and targeted therapies for the prevention and clinical management of periodontitis (Fig. 2).

## HO-1 targeted drug therapy for periodontitis

The present study demonstrated a robust correlation between oxidative stress and periodontitis, as evidenced by elevated levels of the oxidative damage marker MDA and reduced antioxidant capacity identified in the gingival sulcus fluid (*Lutfioğlu et al., 2017*). Subsequent to systematic scaling and root planing interventions, reduced levels of reactive oxygen species and enhanced antioxidant capacity were observed (*Bansal et al., 2017*; *Marconcini et al., 2021*). Combined antibiotic therapy suppresses oxidative stress as well as

improves clinical parameters in periodontitis (*Boia et al., 2019*). Recent studies have revealed that antibiotic treatments like josamycin and tetracycline effectively inhibit periodontitis *via* the HO-1 pathway (*Choi et al., 2018*; *Murakami et al., 2020*). Due to the emergence of antibiotic resistance, there has been a transition towards exploring the utilization of antioxidants for clinical therapy. For instance, the amalgamation of scaling and root planing with antioxidants such as green coffee bean extract oral supplements, lycopene, or green tea extract has demonstrated efficacy in the management of periodontitis (*Saha et al., 2024*; *Tripathi et al., 2019*). For example, a randomized controlled trial of 110 subjects by scholars found that patients with periodontitis had increased clinical attachment levels and reduced gingival inflammation, periodontal pocket depth and oxidative damage after 3 months of topical lycopene application (*Chandra et al., 2012*). Numerous studies have reported significant reductions in probing depth and bleeding index, along with an eight-fold increase in the antioxidant capacity of gingival sulcus fluid following supragingival cleaning and root planing combined with green tea-assisted treatment (*Taleghani et al., 2018*; *Chopra et al., 2016*). In the future, HO-1-targeted antioxidant therapy may represent a significant direction for research, and we will further investigate HO-1-targeted drug development in periodontitis utilizing natural extracts, clinical drugs and novel HO-1 inducers.

## Natural extracts

In recent years, the anti-inflammatory and antioxidant effects of bioactive substances of plant origin as well as herbal medicines have been extensively studied (*Chen et al., 2022b*; *Gu, Hao & Xiao, 2022*). Many natural extracts have been found to exert protective effects on animal models of periodontitis by activating HO-1. HO-1 inducers, such as magnolol, schisandra chinensis α-iso-cubebenol, 6-Shogaol, hesperetin, resveratrol and quercetin, have been investigated for the treatment of periodontitis (*Cho & Kim, 2013*; *Zhao et al., 2021*; *Liu et al., 2019a, 2021*; *Park et al., 2011a*; *Nonaka et al., 2019*; *Bhattarai et al., 2016*).

### Polyphenol compound

Polyphenols are primarily categorized into tannins, flavonoids and lignin-carbohydrate complexes, and are well-known for their antimicrobial, antioxidant and anti-inflammatory properties. In studies pertaining to periodontitis, polyphenols promote collagen synthesis through the activation of fibroblasts, upregulation of collagen gene expression, inhibition of MMPs and antioxidant effects. Furthermore, polyphenols exert anti-inflammatory and antimicrobial effects, as well as inhibit alveolar bone resorption (*Bunte, Hensel & Beikler, 2019*). These mechanisms contribute to the repair, regeneration and maintenance of periodontal tissues, rendering polyphenols valuable therapeutic agents for the treatment of periodontal diseases. Currently, natural extracts of polyphenols, including magnolol, resveratrol, chlorogenic acid, quercetin, hesperetin, isorhamnetin and cardamonin, mediate the inhibition of periodontal inflammation through HO-1 pathways (*Cho & Kim, 2013*; *Huang et al., 2022*; *Liu et al., 2019a, 2021*; *Ma et al., 2024b*; *Jin et al., 2013*; *Okamoto et al., 2024a*). For example, magnolol is a polyphenolic compound extracted from the deciduous tree *Magnolia officinalis*, which belongs to the Magnoliaceae family. Magnolol

exhibits several pharmacological effects, such as anti-inflammatory, antibacterial, anti-ulcer, antioxidant, anticancer and platelet aggregation inhibition. It is mainly used in the treatment of acute enteritis, bacterial or amoebic dysentery, and chronic gastritis. The earliest research on the use of magnolol in periodontal treatment was conducted by *Chang et al. (1998)* who studied the minimum inhibitory concentration (MIC) of magnolol on its antibacterial effect *in vitro*. They found that magnolol had significant antibacterial effects against *Porphyromonas gingivalis*, Actinomyces and *Prevotella intermedia* when the MIC dose was 25 μg/mL (*Chang et al., 1998*; *Cicalău et al., 2021*). Moreover, magnolol could prevent alveolar bone resorption in ligature-induced rat periodontitis by activating the Nrf2/HO-1 pathway, thereby preventing oxidative stress and inflammation (*Lu, Huang & Chou, 2013*; *Lu et al., 2015*). Interestingly, *Liu et al. (2021)* further discovered that magnolol could treat diabetic periodontitis by activating the Nrf2/HO-1 pathway. However, due to the fact that some compounds can regulate different signaling pathways and lack selectivity towards HO-1, many phytochemicals have low bioavailability and may produce certain side effects (*Zhou et al., 2021*). Additionally, resveratrol, a polyphenol found in a variety of foods, has been recognized as a potentially therapeutic drug for the prevention and treatment of inflammatory diseases by targeting and modulating the Nrf2/HO-1 pathway. A clinical study showed that treatment utilizing resveratrol reduced inflammatory markers in the serum and gingival sulcus fluid of patients with periodontitis compared to placebo. However, poor water solubility, rapid decomposition, short serum half-life as well as poor solubility and rapid hepatic and intestinal metabolism result in the low bioavailability of resveratrol (*Zhang et al., 2022*). To solve this problem, researchers grafted resveratrol into mesoporous silica nanoparticles, which improved its bioavailability and allowed for longer drug duration (*Tan et al., 2022*). In addition, there are researchers who have developed a resveratrol delivery system in the form of cyclodextrin and xanthan gum-based oral tablets, which increased the solubility of resveratrol and eliminated bypass enterohepatic metabolism (*Paczkowska-Walendowska et al., 2021*). Moreover, quercetin, a flavonoid compound found in foods such as apples, potatoes, tomatoes and onions, has received much attention for its mediation of the HO-1 pathway and its potent antioxidant and anti-inflammatory properties (*Wei et al., 2021*). However, its functionality is limited due to its water solubility and bioavailability. It was found that combining quercetin with a nanoemulsion increased its absorption rate and eliminated the variability of absorption. The resulting nanoemulsion gel could release 92.4% of quercetin within 6 h (*Aithal et al., 2018*).

*Terpenoid*

Terpenoids are categorized into sesquiterpenes, monoterpenes and diterpenes based on the number of isoprene units they contain. Terpenoids exhibit antimicrobial and anti-inflammatory activities that disrupt microbial biofilms and modulate host immune responses, rendering them valuable in periodontal therapy (*Masyita et al., 2022*). Schisandra is rich in a variety of terpenoids that have been utilized to treat numerous types of inflammation, demonstrating antioxidant, anti-inflammatory, antimicrobial and neuroprotective effects. *Schisandra chinensis* α-iso-cubebenol, extracted from *Schisandra*

*chinensis*, belongs to the sesquiterpenoid group and mediates the PI3K/AKT and ERK pathways to induce HO-1 expression in periodontitis. This process alters the permeability of the outer membrane of periodontally pathogenic bacteria, resulting in microbial destruction while exerting antimicrobial and anti-inflammatory effects (*Park et al., 2011a*).

*Isothiocyanate compounds*

Isothiocyanates represent a class of compounds characterized by the -N=C=S chemical group, formed by substituting sulfur for oxygen in the isocyanate group. Isothiocyanates are known for their antimicrobial, anti-inflammatory, anticancer and antioxidant properties. Among the various isothiocyanates, benzyl isothiocyanate (BITC) is commonly found in cruciferous vegetables such as broccoli, cabbage and bean sprouts (*Rask et al., 2000*). The unique properties of BITC enable it to penetrate the bacterial outer membrane and interfere with the bacterial redox system. Studies have demonstrated its inhibitory activity against certain Gram-negative periodontal pathogens, including Actinomyces and *Porphyromonas gingivalis* (*Sofrata et al., 2011*). Furthermore, several studies have indicated that isothiocyanates, such as sulforaphane, can activate the Nrf2/HO-1 signaling pathway (*Chen et al., 2024*; *Hosokawa et al., 2024*; *Chen et al., 2021b*). Iberin, an isothiocyanate present in green and yellow vegetables, is a specific member of this compound family that exhibits low cytotoxicity to normal cells (*El Badawy et al., 2021*). Research has shown that Iberin can upregulate HO-1 expression and significantly reduce the production of inflammatory mediators in periodontal tissues, including IL-6, CXCL10, VCAM-1, iNOS and COX-2 (*Hosokawa et al., 2022*). Additionally, bertereroin is a bioactive compound classified as an isothiocyanate, primarily found in cruciferous vegetables such as cabbage, arugula and salad greens. Bertereroin has been shown to reduce the production of ROS in periodontal lesions by enhancing the expression of HO-1 (*Hosokawa et al., 2024*).

*Saponin compounds*

The amphiphilic structure of saponins comprises a hydrophilic glycosidic portion and a hydrophobic glycosidic element (saponin backbone), which facilitates the efficient embedding of saponins in the cell membranes of microorganisms. Simultaneously, saponins can interfere with the production of the extracellular matrix necessary for bacterial surface adhesion and biofilm formation (*Khan et al., 2022*). By penetrating the biofilm matrix, saponins induce the death of microbial cells within the structure, thereby disrupting the established biofilm (*Adnan et al., 2023*). Furthermore, saponins derived from various plants exhibit significant antimicrobial, anti-inflammatory and immunomodulatory properties, highlighting their considerable potential for the treatment of periodontal disease. For instance, ginseng is extensively utilized in traditional Chinese medicine to alleviate various diseases, including diabetes, hypertension, gastric ulcers, inflammatory diseases and cancer. Ginseng is characterized by its anticancer, anti-inflammatory, antioxidant and immunomodulatory properties (*Carota et al., 2019*). Research indicates that ginsenosides, the primary pharmacologically active components of ginseng, are closely associated with the regulation of the EGFR/HO-1 pathway. Their

effects on periodontal tissues encompass various aspects, including the regulation of osteoblast and osteoclast functions, inhibition of connective tissue degradation, and exertion of anti-inflammatory, antimicrobial and antioxidant effects (*Kim et al., 2021*).

In summary, plant-derived bioactive compounds and herbs demonstrate significant therapeutic potential in the treatment of periodontitis. Specifically, compounds such as polyphenols, terpenoids, isothiocyanates and saponins significantly influence various aspects of periodontitis, and this effect is closely related to the regulation of HO-1. Moving forward, alongside in-depth studies on the application of these classes of compounds in periodontitis treatment, alkaloids, polysaccharides and quinones are anticipated to function as HO-1 inducers, thereby playing an active role in the prevention and treatment of periodontitis. For example, research has demonstrated that the effectiveness of berberine as an alkaloid, lycium barbarum polysaccharide (LBP) as a polysaccharide and tanshinone IIA as a quinone in various inflammatory diseases is contingent upon the modulatory effect of HO-1 (*Sun et al., 2024*; *Yang et al., 2023*; *Wang et al., 2022a*). Additionally, evidence supports that berberine, LBP and tanshinone IIA may play significant roles in the treatment of periodontitis (*Liu et al., 2019c*; *Qin et al., 2024*; *Wang et al., 2023*; *Mohammadian Haftcheshmeh & Momtazi-Borojeni, 2021*; *Lai et al., 2023*). Consequently, further exploration into whether these alkaloids, polysaccharides and quinones exert their effects *via* the HO-1 pathway will be essential for future studies on the combination and precise medication of multiple compounds. At the same time, future studies should also concentrate on factors such as the route of drug administration, dosage and bioavailability to ensure that these therapeutic approaches are both safer and more effective in clinical applications (Table 1).

### Clinical drugs

Currently, several clinical drugs have been found to activate HO-1 and exhibit potential value in the treatment of periodontitis. For example, nifedipine, a calcium channel blocker widely used in cardiovascular disease treatment, has been reported to possess anti-inflammatory and antioxidant effects. Nifedipine can alleviate osteoarthritis by activating the Nrf2/HO-1 signaling pathway to counteract oxidative stress caused by free radicals (*Yao et al., 2020*). In a mouse macrophage environment stimulated by *Porphyromonas gingivalis* lipopolysaccharide (a periodontal pathogen), nifedipine upregulated HO-1 expression and inhibited the release of inflammatory substances such as NO (*Choe et al., 2021*). Strontium ranelate is an anti-osteoporosis drug with dual effects of promoting bone formation and inhibiting bone resorption (*Yu et al., 2022*). *Souza et al. (2018)* found promotion of HO-1 signaling and inhibition of alveolar bone resorption after ligating the maxillary molars of rats treated with strontium ranelate (100 mg/kg) for 7 days. Dimethyl fumarate (DMF), a drug used for psoriasis and other immune-mediated diseases, weakens intracellular ROS induced by RANKL by enhancing HO-1 expression while inhibiting osteoclastogenesis (*Yamaguchi et al., 2018*; *Matteo et al., 2022*). It is speculated that DMF may be a potential inhibitor of bone destruction in periodontitis. However, the specific efficacy of DMF in periodontitis treatment requires further experimental validation.

**Table 1 Advances *in vitro* therapy with natural extracts.**

| Specific drugs or chemicals | Cell type | Effects on cells | Mechanism | Literatures |
|---|---|---|---|---|
| Chlorogenic acid | HGF | Prevention of LPS-induced inflammatory response | Activation of the Nrf2/HO-1 **anti-inflammatory pathway** | *Huang et al. (2022)* |
| Ginsenosides | HPDL | Promote osteogenic differentiation and inhibit bone resorption, prevent LPS-induced inflammatory response | Promotion of HO-1 **antioxidant and anti-inflammatory pathways** through activation of EGFR | *Kim et al. (2021)* |
| Magnolol | HGF | Prevention of AGEs-induced inflammatory response and apoptosis | Activation of Nrf2/HO-1 **antioxidant and anti-inflammatory pathways** | *Liu et al. (2021)* |
| Panax ginseng fruit | HPDL | Promote osteogenic differentiation and inhibit bone resorption, prevent LPS-induced inflammatory response | Activated HO-1 **antioxidant pathway** | *Kim et al. (2020b)* |
| Resveratrol | HGF | Prevention of LPS-induced inflammatory response | Activation of the Nrf2/HO-1 **antioxidant pathway** | *Bhattarai et al. (2016)* |
|  | PDLSC | Prevention of LPS-induced inflammatory response | Promotion of Nrf2/HO-1 **antioxidant and anti-inflammatory** pathways and inhibition of NF-κB signaling | *Ma et al. (2024b)* |
| Isorhamnetin | HGF | Prevention of LPS-induced inflammatory response | Promotion of Nrf2/HO-1 **antioxidant and anti-inflammatory** pathways and inhibition of NF-κB signaling | *Qi et al. (2018)* |
|  | RAW264. 7 | Prevention of LPS-induced inflammatory response | Promotion of HO-1 **anti-inflammatory pathway** and inhibition of NF-κB signaling | *Jin et al. (2013)* |
| Quercetin | RAW264. 7 | Prevention of LPS-induced inflammatory response | Promotion of HO-1 **anti-inflammatory pathway** and inhibition of STAT1 signaling | *Cho & Kim (2013)* |
| Hesperetin | RAW264. 7 | Inhibition of osteoclast differentiation and activity | Promotion of Nrf2/HO-1 **antioxidant pathway** | *Liu et al. (2019a)* |
| *Schisandra chinensis* α-iso-cubebenol | THP-1 | Prevention of LPS-induced inflammatory response | Promotion of Akt(ERK)/Nrf2/HO-1 **anti-inflammatory pathway**, inhibition of NF-κB signaling | *Park et al. (2011a)* |
| 6-Shogaol | HGF | Prevention of AGEs-induced inflammatory response and apoptosis | Promotion of HO-1 **antioxidant and anti-inflammatory pathways**, inhibition of MAPK and NF-κB signaling | *Nonaka et al. (2019)* |
| Green tea | HGEK | Prevention of LPS-induced inflammatory response | Promotion of Nrf2/HO-1 **antioxidant and anti-inflammatory pathways** | *Hagiu et al. (2020)* |
| Ecklonia cava | RAW264. 7 | Prevention of LPS-induced inflammatory response | Promotion of Nrf2/HO-1 **anti-inflammatory pathway** and inhibition of NF-κB signaling | *Kim et al. (2019)* |
| Ginkgo biloba | RAW264. 7 | Prevention of LPS-induced inflammatory response | Promoting the Nrf2/HO-1 **anti-inflammatory pathway** | *Ryu et al. (2012)* |
| Sappanchalcone | HPDL | Prevention of LPS-induced inflammatory response | Promoting the JNK/HO-1 **anti-inflammatory pathway** | *Jeong et al., 2010)* |
| Kaempferol | RAW264. 7 | Prevention of LPS-induced inflammatory response | Promoting HO-1 **antioxidant and anti-inflammatory pathways** | *Choi et al. (2013)* |
| Cardamonin | HPDL | Prevention of IL-1β-induced inflammatory response | Promoting HO-1 **anti-inflammatory pathway**, inhibiting NF-κB and STAT3 signalling | *Okamoto et al. (2024b)* |
| Xanthones | PDLSC | Prevention of H$_2$O$_2$-induced inflammation, promotion of osteogenic differentiation | Promoting the Nrf2/HO-1 **antioxidant pathway** | *Ruangsawasdi et al. (2023)* |
| Notopterol | HGF | Prevention of LPS-induced inflammatory response | Inhibition of NF-κB and promotion of the PI3K/AKT/HO-1 signalling pathway | *Zhou et al. (2023)* |

(Continued)

| Specific drugs or chemicals | Cell type | Effects on cells | Mechanism | Literatures |
|---|---|---|---|---|
| Berteroin | HPDL | Prevention of IL-1 β or TNF-α induced inflammatory response | Promoting HO-1 **anti-inflammatory pathway**, inhibiting NF-κB and STAT3 signalling | *Hosokawa et al. (2024)* |
| curcumin | H400 oral epithelial cell | Prevention of Fusobacterium nucleatum-induced inflammatory response | Promoting the HO-1/CO **anti-inflammatory pathway** | *Grant et al. (2023)* |
| Iberin | TR146 cells | Prevention of TNF-α-induced inflammatory response | Promoting HO-1 **anti-inflammatory pathway**, inhibiting NF-κB, STAT 3 and S6 signalling | *Hosokawa et al. (2022)* |

**Note:**
HGFs, human gingival fibroblasts; HPDL, human periodontal ligament cells; THP-1, human monocytes; HGEK, human gingival epithelial keratin-forming cells; PDLSC, periodontal stem cells; TR146, a human oral epithelial cell; AGEs, advanced glycosylation end products; LPS, lipopolysaccharides; EGFR, epidermal growth factor receptor.

Additionally, commonly used hypoglycemic agents for diabetes, a prevalent oxidative stress disorder, include biguanides (*e.g.*, metformin) and GLP-1 (glucagon-like peptide-1) receptor agonists. Metformin is regarded as the first-line choice for treating type 2 diabetes, effectively reducing intracellular free radical levels, enhancing insulin sensitivity, and exerting a cytoprotective effect through the upregulation of HO-1 expression (*Wang et al., 2022b*). Conversely, GLP-1 receptor agonists (*e.g.*, liraglutide) represent an emerging class of antidiabetic drugs that stimulate insulin secretion primarily by mimicking the biological effects of GLP-1. Research has demonstrated that GLP-1 receptor agonists induce HO-1 expression, promote endothelial cell protection and angiogenesis, contribute to diabetic wound healing, and reduce the risk of diabetes-related complications, thereby enhancing the overall prognosis of diabetic patients (*Huang et al., 2021c*). The pathogenesis of periodontitis, an oxidative stress disorder closely associated with diabetes, involves the interplay between oxidative stress and inflammatory responses. Numerous studies have demonstrated that both biguanides and GLP-1 receptor agonists effectively inhibit the development and progression of periodontitis (*Pang et al., 2019*; *Sawada et al., 2020*; *Zhang et al., 2020*; *Neves et al., 2023*). Consequently, an in-depth investigation into the specific role of HO-1 in the pathogenesis of periodontitis concerning biguanides and GLP-1 receptor agonists is essential for future treatment and management strategies.

Furthermore, atherosclerosis is an oxidative stress disorder primarily characterized by lipid accumulation within the vessel wall. Research indicates that statins effectively reduce oxidative stress and enhance endothelial function by increasing HO-1 expression levels and inhibiting NADPH oxidase activity, thereby minimizing vascular damage and preserving vascular structural integrity (*Piechota-Polanczyk & Jozkowicz, 2017*). In a randomized controlled trial concerning periodontitis, scaling and root planing combined with adjunctive statin treatment significantly improved clinical attachment levels and reduced periodontal pocket depth, while effectively decreasing the incidence of bleeding on probing (*Alkakhan et al., 2023*). Recent studies further suggest that statins possess significant potential for periodontal regeneration, osteoclast modulation and antimicrobial effects in periodontitis (*de Carvalho et al., 2021*; *Parolina de Carvalho et al., 2024*). An in-depth study of the mechanism of action of statins in periodontitis, especially the way

**Table 2 Advances in clinical drug therapy *in vitro*.**

| Specific drugs or chemicals | Cell type | Effects on cells | Mechanism | Literatures |
|---|---|---|---|---|
| Nifedipine | RAW264. 7 Cell | Prevention of LPS-induced inflammatory response | Promotes HO-1 **anti-inflammatory pathway**, inhibits NF-κB and STAT 1/3 signaling, and promotes M2 macrophage polarization | *Yao et al. (2020)* |
| Josamycin | RAW264. 7 Cell | Prevention of LPS-induced inflammatory response | Promotion of HO-1 **anti-inflammatory pathway** and inhibition of NF-κB signaling | *Choi et al. (2018)* |
| Telmisartan | RAW264. 7 Cell | Prevention of LPS-induced inflammatory response | Promotes HO-1 **anti-inflammatory pathway**, inhibits NF-κB and STAT 1/3 signaling, and promotes M2 macrophage polarization | *Choe et al. (2019)* |
| Dimethyl fumarate | RAW264. 7 Cell | Inhibits osteoclast formation and activity | **Inhibition of oxidative stress** *via* the Nrf2/HO-1 pathway | *Yamaguchi et al. (2018)* |
| Tetracycline | RAW264. 7 Cell | Prevention of LPS-induced inflammatory response | Promoting HO-1 **antioxidant and anti-inflammatory pathways** | *Murakami et al. (2020)* |

they exert their effects through the HO-1 pathway, will bring important insights and guidance for the future treatment and management of periodontitis.

In summary, the management of oxidative stress disorders is closely linked to the function of HO-1. Consequently, a thorough investigation into the specific applications of these HO-1-targeted drugs in periodontitis will be vital for future research endeavors. This approach will not only facilitate the enhancement of combination drug strategies for patients with concomitant systemic oxidative stress disorders, thereby avoiding unnecessary duplication of therapy, but is also expected to achieve optimal efficacy and a favorable prognosis. From the perspective of drug regulatory approvals and safety studies, the available clinical drugs that act as HO-1 inducers demonstrate the potential to become effective options for the treatment of periodontitis (Table 2).

### Novel HO-1 inducers

In addition to the use of plants, herbs and clinical drugs for HO-1 induction, there has been a recent interest in other modes of obtaining HO-1 inducers for the treatment of periodontitis. Carbon monoxide has been shown to play a significant role in various cellular biological processes, including cell apoptosis and immune-inflammatory responses. Carbon monoxide-releasing molecules (CORMs), as a novel type of compound, can release carbon monoxide in a controlled manner under physiological conditions, thereby increasing the expression of HO-1 in various animal models and cell types (*Lv et al., 2020*). *Choi et al. (2021, 2022, 2015a)* demonstrated that lipophilic CORM-2 and water-soluble CORM-3, CORM-401 had inhibitory effects on the inflammatory response induced by *Porphyromonas gingivalis* lipopolysaccharide through NF-κB inhibition. Additionally, it has been reported that CORM-3 can suppress the expression of the osteoclast factor RANKL and upregulate the expression of osteoprotegerin (OPG), which may have potential therapeutic value in inhibiting bone resorption in periodontitis (*Lv et al., 2020*). Moreover, vitamin D has been found to have regulatory effects in various inflammatory diseases. ED-71, as an analog of vitamin D, can activate the Nrf2/HO-1 pathway and inhibit the expression of NLRP3, Caspase-1 and IL-1β in gingival fibroblasts

under inflammatory conditions, thereby suppressing pyroptosis (*Huang et al., 2020*). Interestingly, coffee is a common dietary ingredient in everyday life, and opinions differ as to whether it is beneficial or harmful to health. A survey of 16, 730 adults in Korea found that increased coffee intake may further predispose to periodontitis (*Han, Hwang & Park, 2016*). However, a Japanese cross-sectional study found that coffee consumption during periodontal maintenance treatment was associated with a reduction in the prevalence of severe periodontitis (*Machida et al., 2014*). As coffee is a complex mixture, different types of coffee have different compositions and different biological effects. This is perhaps why the results of current coffee research are inconsistent (*Song et al., 2022*). The main components of coffee are caffeine and chlorogenic acid, which are absorbed by the body. Caffeine and chlorogenic acid have been reported to promote the expression of Nrf2 translocation and HO-1, showing antioxidant effect (*Huang et al., 2022*; *Khan et al., 2019*). In contrast, a recent study showed that caffeine and chlorogenic acid preparation of artificial coffee promoted AMP-activated protein kinase phosphorylation and reduced the NF-κB pathway to exert anti-inflammatory effects on periodontal cells (*Song et al., 2022*). However, based on the uncertainty that still exists regarding the toxicity of these synthetic as well as dietary-acquired substances, and what constitutes a safe and effective dosage. Further studies need to be conducted before the substances are ultimately converted into clinical drugs (Tables 3 and 4).

In summary, natural extracts exhibit a reduced incidence of side effects and a lower propensity for resistance compared to their chemical counterparts. Nevertheless, they are confronted with challenges related to extraction purity, potency, and dosage. Polyphenols, terpenoids, isothiocyanates and saponins have been shown to significantly influence various facets of periodontitis *via* the modulation of HO-1 activity. These compounds have been extensively investigated within the domain of periodontitis. Future research should prioritize the exploration of additional polyphenols, terpenoids, isothiocyanates and saponins, as well as the translation of experimental findings into clinical applications. Furthermore, alkaloids, polysaccharides and quinones exhibit a strong correlation with HO-1, oxidative stress disorders and chronic inflammatory diseases. Nonetheless, there exists a paucity of studies addressing the principal pathways of these compound classes in periodontitis, including HO-1, warranting our sustained long-term research efforts. The adverse effects associated with marketed clinical drugs are more explicitly delineated, and their safety and efficacy are thoroughly documented through clinical trials. The HO-1-targeted clinical drugs employed in the treatment of oxidative stress-related diseases represent promising candidates for the management of periodontitis. Researchers have initiated investigations into novel HO-1 inducers, including synthetic CORM, periostin and compounds derived from dietary vitamins and coffee. However, these innovative HO-1 inducers presently lack adequate clinical evidence to substantiate their efficacy. Consequently, a significant focus of future research will involve the application of acquired clinical data to the development of clinical therapeutics for periodontal disease. Furthermore, it is crucial to persist in investigating the role of HO-1 in periodontitis management as mediated by natural extracts and clinical drugs. This ongoing research will facilitate the development of diverse structures and functional groups to enhance HO-1

**Table 3 Advances *in vitro* therapy with novel HO-1 inducers.**

| Specific drugs or chemicals | Cell type | Effects on cells | Mechanism | Literatures |
|---|---|---|---|---|
| CORM-2 | RAW264. 7 | Prevention of LPS-induced inflammatory response | Promotion of HO-1/CO **anti-inflammatory pathway**, inhibition of NF-κB and STAT 1/3 signaling | *Choi et al. (2021)* |
| CORM-3 | HPDL | Prevention of nicotine/LPS-induced inflammatory responses and inhibition of osteoclast formation and activity | Promoting HO-1/CO pathway **antioxidant and anti-inflammatory** pathway | *Song et al. (2017)* |
| CORM-401 | RAW264. 7 | Prevention of LPS-induced inflammatory response | Promotion of HO-1/CO **anti-inflammatory pathway**, inhibition of NF-κB pathway | *Choi et al. (2022)* |
| Caffeine | Oral keratinocytes | Prevention of LPS-induced inflammatory response | Promotion of Nrf2/HO-1 **antioxidant and anti-inflammatory pathways** | *Song et al. (2022)* |
| DHA | RAW264. 7 | Prevention of LPS-induced inflammatory response | Promotion of HO-1 **anti-inflammatory pathway**, inhibition of NF-κB and JNK1/2 pathways | *Choi et al. (2014)* |
| Caffeic acid phenethyl ester | THP-1 | Prevention of saliva-induced inflammatory response in patients with periodontitis | Promotion of HO-1 **anti-inflammatory pathway**, inhibition of NF-κB pathway | *Huang et al. (2021b)* |
| | RAW264. 7 | Prevention of LPS-induced inflammatory response | Promotion of HO-1 **anti-inflammatory pathway**, inhibition of SOCS1, inhibition of NF-κB and STAT 1/3 signaling | *Choi et al. (2015b)* |
| Periostin | PDLF | Prevention of LPS-induced inflammatory response and apoptosis | Promotion of Nrf2/HO-1 **antioxidant and anti-inflammatory pathways** | *Jiang et al. (2022)* |
| Surfactin | HGFs, THP-1 | Prevention of PM-induced inflammatory response | Promotion of Nrf2/HO-1 **antioxidant and anti-inflammatory pathway**, inhibition of VCAM-1-dependent pathway | *Vo et al. (2022)* |
| ED-71 | HGFs | Prevention of LPS-induced cellular pyroptosis | Promotion of Nrf2/HO-1 **antioxidant pathway**, inhibition of NLRP3 inflammatory vesicles | *Huang et al. (2020)* |
| Melatonin-derived carbon dots | RAW264. 7 | Prevention of $H_2O_2$-induced inflammatory response | Promoting Nrf2/HO-1 **antioxidant** and **anti-inflammatory pathways** | *Xin et al. (2024)* |
| BML-111 | PDLF | Prevention of $H_2O_2$-induced cellular pyroptosis and osteogenic dysfunction | Promoting Nrf2/HO-1 **antioxidant and anti-inflammatory pathways** | *Xu et al. (2024)* |
| Nitro-conjugated linoleic acid | Raw 264. 7 | Prevention of LPS-induced inflammatory response | Promoting HO-1 **anti-inflammatory pathway**, inhibiting NF-κB and STAT 1/3 signalling, and promoting M2 macrophage polarisation | *Lee et al. (2024)* |
| Glutathione Peroxidase-Mimicking Nanozyme | PDLSC | Prevention of $H_2O_2$-induced inflammation, promotion of osteogenic differentiation | Promotion of Nrf2/HO-1 **antioxidant pathway**, promotion of PI3K/Akt signalling | *Zhu et al. (2024)* |
| vitamin D | HGF | Prevention of AGEs-induced inflammatory response | Promoting Nrf2/HO-1 **antioxidant and anti-inflammatory pathways** | *Lu et al. (2023)* |
| Cl-amidine | HGF | Prevention of LPS-induced inflammatory response | Promoting Nrf2/HO-1 **antioxidant and anti-inflammatory pathway**, inhibiting NF-κB and JNK/MAPK signaling | *Du et al. (2022)* |

**Note:**
CORM, carbon dioxide releasing molecule; ED-71, vitamin D analog; PDLF, periodontal fibroblasts; PM, particulate matter; SOCS1, suppressor of cytokine signaling; VCAM-1, vascular cell adhesion molecule-1.

**Table 4 Advances *in vivo* pharmacotherapy of HO-1-induced periodontitis.**

| Types of HO-1 inducers | Specific drugs or chemicals | Mouse type/ human | Effects on mice | Literatures |
|---|---|---|---|---|
| Natural extracts | Chlorogenic acid | Male C57BL/6 mice | Chlorogenic acid-loaded nanomicelles lead to reduced inflammation and inhibited bone resorption in a simulated experimental periodontitis model | *Li et al. (2022)* |
| | Magnolol | Male Sprague Dawley rats | Reduction of inflammation and inhibition of bone resorption in a simulated experimental periodontitis model | *Lu, Huang & Chou (2013)* |
| | Quercetin | Male C57BL/6 mice | Elevated Nrf2 expression in a simulated experimental periodontitis model leads to reduced oxidative damage, reduced inflammation, and inhibition of bone resorption | *Wei et al. (2021)* |
| | | Male wistar rats | Reduction of inflammation and inhibition of bone resorption in a simulated experimental periodontitis model | *Taskan & Gevrek (2020)* |
| | 6-Shogaol | Male C57BL/6 mice | Reduction of inflammation and inhibition of bone resorption in a simulated experimental periodontitis model | *Kim et al. (2020a)* |
| | Green tea | Male C57BL/6 mice | Reduction of inflammation and inhibition of bone resorption in a simulated experimental periodontitis model | *Kaboosaya et al. (2020)* |
| | Ginkgo biloba | Male Wistar rats | Inhibition of bone resorption in a simulated experimental periodontitis model | *Sezer et al. (2013)* |
| | Kaempferol | Wistar rats | Reduction of alveolar bone resorption, attachment loss, and MMP-1 and 8 production in experimental periodontitis. | *Balli et al. (2016)* |
| | Mono-carbonyl analogues of curcumin (MCACs) | Male Sprague Dawley rats | Reduction of inflammation and inhibition of bone resorption in a simulated experimental periodontitis model | *Zhao et al. (2021)* |
| | Panax ginseng fruit | Sprague Dawley rats | Reduced inflammation in a simulated experimental periodontitis model | *Kim et al. (2020b)* |
| | Resveratrol | Male Sprague Dawley rats | Reduction of inflammation and inhibition of bone resorption in a simulated experimental periodontitis model | *Bhattarai et al. (2016)* |
| | Ecklonia cava | Male Sprague Dawley rats | Reduction of inflammation and inhibition of bone resorption in a simulated experimental periodontitis model | *Kim et al. (2019)* |
| | Epigallocatechin-3-gallate | Sprague Dawley rats | Reduced inflammation, antioxidants, and inhibition of bone resorption in a simulated experimental periodontitis model | *Fan et al. (2023)* |
| Clinical drugs | Nifedipine | male BALB/c mice | Inhibition of bone resorption in a simulated experimental periodontitis model | *Lee et al. (2023b)* |
| | Strontium ranelate | Male Wistar rats | Inhibition of bone resorption in a simulated experimental periodontitis model | *Souza et al. (2018)* |
| | Dimethyl fumarate | male BALB/c mice | Inhibition of bone resorption in a simulated experimental periodontitis model | *Yamaguchi et al. (2018)* |
| | Tetracycline | Human | Reduction of periodontal pocket depth and clinical attachment levels in patients with periodontitis | *Sinha et al. (2014)* |

| Types of HO-1 inducers | Specific drugs or chemicals | Mouse type/ human | Effects on mice | Literatures |
|---|---|---|---|---|
| Novel HO-1 inducers | CORM-2 | Wistar rats | Reduced inflammation in a simulated experimental periodontitis model | *Hou et al. (2014)* |
| | caffeine | Male Wistar rats | Antimicrobial action, promotion of alveolar bone repair in a simulated experimental periodontitis model | *Sari et al. (2023)* |
| | caffeic acid phenethyl ester | Male Wistar rats | Reduced inflammation, antioxidants, and inhibition of bone resorption in a simulated experimental periodontitis model | *Yiğit et al. (2021)* |
| | Melatonin-derived carbon dots | Male C57BL/6 mice | Reduced inflammation, antioxidants, and inhibition of bone resorption in a simulated experimental periodontitis model | *Xin et al. (2024)* |
| | Ixeris dentata and *Lactobacillus gasseri* media | Male C57BL/6 mice | Reduced inflammation, antioxidants, and inhibition of bone resorption in a simulated experimental periodontitis model | *Lee et al. (2023a)* |

expression and the treatment of periodontitis, ultimately leading to the innovation of novel compounds aimed at achieving a comprehensive cure for this condition.

## CONCLUSION

In recent years, researchers have made significant strides in elucidating the pathogenesis of periodontitis through a comprehensive analysis of the underlying mechanisms of oxidative stress involved in this condition. This review delineates the multifaceted functions of HO-1, encompassing its roles in antioxidant activity, anti-apoptosis, anti-pyroptosis, anti-inflammatory responses and regulation of bone homeostasis, positing HO-1 as a promising novel target for the development of therapies for periodontitis. Numerous studies have demonstrated that natural extracts, clinical drugs and novel HO-1 inducers have investigated the potential of targeting periodontitis through the augmentation of HO-1 expression. Nevertheless, despite notable progress, the therapeutic potential of HO-1 has yet to be fully realized within a clinical context.

Firstly, existing investigations on periodontitis predominantly remain at the cellular and animal experiment stages, necessitating further validation through clinical trials. Secondly, the dualistic role of HO-1 in the treatment of periodontitis warrants attention. Sustained overexpression of HO-1 may result in the accumulation of heme degradation products, which not only proves detrimental to the treatment of periodontitis but may also enhance cytotoxicity. Therefore, optimizing therapeutic efficacy while ensuring the safety of HO-1-targeted therapies represents a crucial avenue for future research. Thirdly, to thoroughly investigate the role of polyphenols, terpenoids, isothiocyanates, saponins, alkaloids, polysaccharides and quinones in the HO-1-related pathway associated with periodontitis, it is crucial to examine the interrelationships between various functional groups, HO-1 and periodontitis. This study aims to elucidate the potential of these compounds in the treatment of periodontitis, thereby offering innovative insights and opportunities for the future development of novel HO-1 compounds. Furthermore, investigating the effects of HO-1 targeted drugs related to oxidative stress diseases on the development and

progression of periodontitis may mitigate the risk of medication duplication in patients suffering from systemic oxidative stress diseases concomitant with periodontitis, thereby facilitating more precise therapeutic interventions for optimal efficacy and prognosis. Currently, the antioxidant and anti-inflammatory regulatory mechanisms associated with the HO-1 signaling pathway remain to be fully elucidated. Future investigations should delve into the involvement of HO-1 in the inter-regulation of critical pathways, including MAPK, NF-κB and PI3K/Akt, as well as the potential of HMGB1 as a significant target for the regulation of periodontitis treatment by HO-1. Identifying drugs that target HO-1-related microRNAs or downstream target genes associated with HO-1 may offer novel insights for the treatment of periodontitis.

In conclusion, HO-1 serves as a potentially crucial anti-inflammatory, antioxidant, immunomodulator and bone modulator in the context of periodontitis, necessitating comprehensive exploration to ascertain the bioavailability and safety of HO-1-targeted therapeutic agents. Moving forward, it is essential to elucidate the regulatory mechanisms of HO-1 as a potent antioxidant, address the limitations of HO-1-targeted therapies for periodontitis treatment, and ultimately achieve the clinical application of these drugs as innovative therapeutic agents for periodontitis.

## ACKNOWLEDGEMENTS

We thank Dr Xiaoxiao Pang for his invaluable help in editing the manuscript.

### Funding

This work was supported by the Nanchong City-School Science and Technology Strategic Cooperation Project (NSMC20170301), the Natural Science Foundation of Science and Technology Department of Sichuan Province(2023NSFSC0646), the Research Development Fund of the Affiliated Hospital of North Sichuan Medical College (No.2023PTZK006), the Youth Program of the Affiliated Hospital of North Sichuan Medical College (CBY22-QNA65), the Research and Development Program of the Affiliated Hospital of North Sichuan Medical College (2023JC048), the International Student Teaching Project of North Sichuan Medical College (ISJG2023-14), and the Open Lab Program of North Sichuan Medical College (PGJ2022004). The funders had no role in study design, data collection and analysis, decision to publish, or preparation of the manuscript.

### Grant Disclosures

The following grant information was disclosed by the authors:
Nanchong City-School Science and Technology Strategic Cooperation Project: NSMC20170301.
Natural Science Foundation of Science and Technology Department of Sichuan Province: 2023NSFSC0646.
Research Development Fund of the Affiliated Hospital of North Sichuan Medical College: 2023PTZK006.

Youth Program of the Affiliated Hospital of North Sichuan Medical College: CBY22-QNA65.

Research and Development Program of the Affiliated Hospital of North Sichuan Medical College: 2023JC048.

International Student Teaching Project of North Sichuan Medical College: ISJG2023-14.

Open Lab Program of North Sichuan Medical College: PGJ2022004.

## Competing Interests

The authors declare that they have no competing interests.

## Author Contributions

- Weiwei Lv conceived and designed the experiments, performed the experiments, analyzed the data, prepared figures and/or tables, and approved the final draft.
- Shichen Hu conceived and designed the experiments, performed the experiments, analyzed the data, prepared figures and/or tables, and approved the final draft.
- Fei Yang performed the experiments, prepared figures and/or tables, and approved the final draft.
- Dong Lin performed the experiments, prepared figures and/or tables, and approved the final draft.
- Haodong Zou performed the experiments, prepared figures and/or tables, and approved the final draft.
- Wanyan Zhang performed the experiments, authored or reviewed drafts of the article, and approved the final draft.
- Qin Yang performed the experiments, authored or reviewed drafts of the article, and approved the final draft.
- Lihua Li analyzed the data, authored or reviewed drafts of the article, and approved the final draft.
- Xiaowen Chen analyzed the data, authored or reviewed drafts of the article, and approved the final draft.
- Yan Wu conceived and designed the experiments, analyzed the data, authored or reviewed drafts of the article, and approved the final draft.

## Data Availability

This is a literature review.

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
