# Peer review of "Heme oxygenase-1: potential therapeutic targets for periodontitis"

_PeerJ, doi:10.7717/peerj.18237_

## Round 0.1 · original submission · Major Revisions

Dear Authors, please revise the MS carefully according to reviewers' comments

Reviewer 1 ·

Basic reporting

1. This manuscript is a theoretical review of the enzyme heme oxygenase (HO-1) which is thought to function as an ROS inhibitor in dental gum damage as a result of plaque on teeth and gums. This paper is useful for the treatment of tooth decay and the drugs that will be produced to treat decay or tooth loss.
2. Quite a number of references were cited in preparing this manuscript. The majority of references in preparing this manuscript are sourced from quality papers indexed by Scopus, WoS, and PubMed.
3. Introduction
The heme oygenase (HO) enzyme has 2 isoforms, namely HO-2 and HO-3. Why did the author choose HO-1 as the candidate? Both function to induce responses to various oxidative stressors. Is it likely that both isoforms of heme oxygenase play an important role in protecting cells and tissues from damage by breaking down heme and producing products with antioxidant and anti-inflammatory properties. These are important enzymes in the body's defense against various pathological conditions.
It would have been better if the author had also mentioned the other isoforms at the beginning of this paper, so that the reader can better understand why the author emphasizes the HO-1 enzyme.
HO-1 metabolite
HO-1 is effective when there is degradation of heme by HO-1. There are 3 metabolites produced due to the degradation of free heme by HO-1 that produce by-products such as CO, Fe2+, and Biliverdin and Bilirubin.
Periodontal cell metabolism cannot be achieved without the help of heme, but excessive heme will catalyze the production of ROS which can cause oxidative stress and cell damage. The three metabolites can act as cell protection. Based on this brief description, the author should add a sentence regarding the performance of the HO-1 enzyme usually occurs because periodontitis has already occurred or symptoms of periodontitis are just starting to appear?

Carbon Monoxide (CO)
CO comes from heme degradation by HO-1, the role of CO is very important in overcoming periodontitis, it should also be described how CO is available from heme degradation?

Fe2+
It is suggested that the authors add how Fe2+ is formed from heme and under what conditions?
HO-1 in periodontal tissues
Many studies have shown that HO-1 is expressed at low levels in normal periodontal tissues, and only a small amount is secreted to maintain the balance between oxidative and antioxidant status in the cell.
Can a normal periodontal tissue form be shown in addition to this paper?

HO-1 and periodontitis
Pathophysiological mechanism

Good explanation by the author in this section. OK

Molecular mechanism
OK

HO-1 Targeted Drug Therapy for Periodontitis
”.... Recent studies have 337 revealed that antibiotic treatments such as crossamycin and tetracycline effectively inhibit 338 periodontitis via the HO-1 pathway[103, 104]..."
This part is interesting. How both forms of antibiotics are able to inhibit periodontitis via the HO-1 pathway. Can the authors add to this?
"Due to the emergence of antibiotic resistance, 339 there has been a transition towards exploring the utilization of antioxidants for clinical therapy. For example, the amalgamation of scaling and root planing with antioxidants such as green coffee bean extract oral supplements, lycopene, or green tea extract has demonstrated efficacy in the management of periodontitis[105, 106]”.
In application, the use of green coffee bean extract, lycopene, or green tea is efficacious in the management of periodontitis. Is there any data that can be presented in this paper that shows this efficacy?
Natural extracts
OK
Clinical drugs
OK
Small molecule compounds
“Further research should be focused on translating the obtained clinical data into the development of clinical drugs for periodontal diseases”.
Is this the only one suitable for research in the future?
In this paper, there is something missing about the indicators in the observation of periodontitis that have been carried out by previous researchers. For example, how to keep the concentration of CO and Fe2+ low.
The methodological aspect of each edited article is very important in a manuscript like this, as it is based on a collection of research articles on a scientific field.
4. Please paraphrase this article so that the level of similarity with similar published articles can be minimized.
5. Thank you

Experimental design

no comment

Validity of the findings

no comment

Additional comments

This manuscript is good enough, only needs additional explanation in writing and references, please look again for the latest one

Annotated reviews are not available for download in order to protect the identity of reviewers who chose to remain anonymous.

Reviewer 2 ·

Basic reporting

Dear Authors,
This review article explores the potential role of heme oxygenase-1 (HO-1) as a therapeutic target for periodontitis. While the topic is timely and relevant, the manuscript requires substantial revisions to improve clarity, scientific rigor, and adherence to principles of scientific writing.
Title and Abstract:
• The title "HO-1: Potential therapeutic targets for periodontitis" is concise but could be more specific. Consider revising to "Heme oxygenase-1 as a potential therapeutic target for periodontitis" to provide more clarity.
• Lines 25-26: The phrase "urgent requirement for suitable and effective antioxidant therapies" seems overstated. Consider rephrasing to "need for improved antioxidant therapies."
• Lines 34-37: The conclusion could be more specific about the potential applications of HO-1-targeted therapy in periodontitis.
Introduction:
• Line 40-44: The introduction should more clearly define the scope and significance of the study. Consider elaborating on the impact of periodontitis globally and the current limitations in treatment (https://doi.org/10.1111/jcpe.12732).
• Line 44: You could add a sentence here referencing the immune cell aspects covered in the suggested paper, for example: "The complex interplay between innate and adaptive immune cells also plays a crucial role in the pathogenesis and progression of periodontitis [https://doi.org/10.1111/odi.14360]. In particular, oxidative stress is a key factor during the development of periodontitis.
• Line 46-47: The example of LncRNAZFY-AS1 seems abrupt. A smoother transition is needed to introduce this concept.
• Lines 57-59: This sentence could be more concise. For example: "Oxidative stress, particularly reactive oxygen species (ROS), plays a crucial role in the pathogenesis of periodontitis."
• Line 57-63: The role of ROS in periodontal disease is crucial but could be explained in simpler terms for broader understanding.
• Lines 79-84: This paragraph could benefit from a clearer structure, perhaps grouping findings by type of antioxidant marker.
• Lines 90-105:The paragraph on HO-1 lacks detail and context. Expand on the different isoforms, their functions, and their relevance to periodontitis (https://doi.org/10.1146/annurev.pharmtox.010909.105600).

Experimental design

Methods:
• Lines 107-112: The methodology for the literature search is very brief. Including more details such as the date range of articles considered, inclusion/exclusion criteria, and the number of articles reviewed, would be helpful.
• The list of keywords is extensive and includes terms not directly related to HO-1. Refine the list to focus on the central theme.

Validity of the findings

Results and discussion:
• Line 114: The review would benefit from a more structured approach, perhaps with clearer subheadings to guide the reader through different aspects of HO-1 concerning periodontitis.
• Lines 115-124: This section provides a good overview of heme and its role. However, it could benefit from a brief explanation of how heme relates specifically to periodontitis.
• Lines 143-153: The discussion of Fe2+ is confusing. Clarify the dual role of iron ions in lipid peroxidation and provide a more balanced analysis of their potential effects in periodontitis.
• Lines 177-212: The manuscript often relies on broad statements and lacks in-depth analysis of specific studies and their findings. For instance, the discussion on the role of HO-1 in different periodontal tissues would benefit from a more detailed analysis of individual cell types and their specific responses to HO-1 modulation.
• The CO, Fe2+, and Biliverdin/Bilirubin subsections are informative but could focus more on their relevance to periodontitis.
• The potential therapeutic applications of HO-1 in periodontitis could be expanded upon, as this is a key aspect mentioned in the title and abstract (.
• Lines 215-328: This section is dense and difficult to follow. Break it down into smaller paragraphs with clear subheadings to improve readability.
• Lines 271-328: The discussion of the molecular mechanisms is particularly convoluted. Simplify the language, use shorter sentences, and provide a more structured analysis of the signaling pathways.
• The limitations and future research directions are appropriately acknowledged. Elaborating more on the specific challenges to clinical translation and how they might be overcome would further strengthen this section (https://doi.org/10.1016/j.archoralbio.2017.12.022).
• The figure captions for Figure 1 and Figure 2 are too long and repetitive. Condense them to concise descriptions of the figures' key elements.
• Lines 347-386: The discussion of natural extracts lacks a systematic approach. Organize the information by specific compounds and provide a more detailed analysis of their mechanisms of action, efficacy, and limitations.
• Lines 388-404: The discussion of clinical drugs is limited to a few examples. Expand on other potential candidates and their current status in periodontitis research.
Conclusions:
• Line 465-471: The conclusion should more explicitly identify gaps in the current research and suggest specific future research directions.

Additional comments

General Comments:
• The manuscript requires editing for grammatical errors and clarity. A thorough language and style revision is necessary.
• Overall, this manuscript can potentially be a valuable contribution to the field. However, significant revisions are required to improve clarity, scientific rigor, and adherence to principles of scientific writing. I encourage the authors to consider the comments and suggestions carefully and revise the manuscript accordingly.

---

## Round 0.2 · accepted · Accept

Now the manuscript can be accepted.

Reviewer 1 ·

Basic reporting

Comment to Author
1. Writing should be consistent within the manuscript for example “NF-KB”. Please check on your manuscript.
2. Double-check the spacing issues in the manuscript writing.
3. Double check the requirements according to the PeerJ template
4. Use of Font size and Font Type must be in accordance with the rules in PeerJ .
5. The content of the manuscript is adequate and the author needs to pay attention to points 1-3.
6. Minor improvement as the writer needs to rephrase the manuscript. The content and substance of the writing is good.
7. What I questioned in the initial review has been explained by the author.

Experimental design

no comment

Validity of the findings

no comment

Additional comments

see after plagiarism check by Turnitin. The manuscript is OK and only 10%.

Annotated reviews are not available for download in order to protect the identity of reviewers who chose to remain anonymous.

Reviewer 2 ·

Basic reporting

Overall, the authors have successfully addressed the majority of the reviewers' comments with a high level of detail and precision. The manuscript has been significantly improved in terms of structure, clarity, and content.

Experimental design

the manuscript is well-written and addresses the majority of the requested changes.

Validity of the findings

The findings presented in the manuscript are generally valid and well-supported by existing literature.